# Delayed Chromosome Alignment to the Spindle Equator Increases the Rate of Chromosome Missegregation in Cancer Cell Lines

**DOI:** 10.3390/biom9010010

**Published:** 2018-12-28

**Authors:** Kinue Kuniyasu, Kenji Iemura, Kozo Tanaka

**Affiliations:** Department of Molecular Oncology, Institute of Development, Aging and Cancer, Tohoku University, 4-1 Seiryo-machi, Aoba-ku, Sendai, Miyagi 980-8575, Japan; kinue.kuniyasu.b1@tohoku.ac.jp (K.K.); kenji.iemura.a6@tohoku.ac.jp (K.I.)

**Keywords:** chromosome segregation, mitosis, chromosomal instability, molecular motors

## Abstract

For appropriate chromosome segregation, kinetochores on sister chromatids have to attach to microtubules from opposite spindle poles (bi-orientation). Chromosome alignment at the spindle equator, referred to as congression, can occur through the attachment of kinetochores to the lateral surface of spindle microtubules, facilitating bi-orientation establishment. However, the contribution of this phenomenon to mitotic fidelity has not been clarified yet. Here, we addressed whether delayed chromosome alignment to the spindle equator increases the rate of chromosome missegregation. Cancer cell lines depleted of Kid, a chromokinesin involved in chromosome congression, showed chromosome alignment with a slight delay, and increased frequency of lagging chromosomes. Delayed chromosome alignment concomitant with an increased rate of lagging chromosomes was also seen in cells depleted of kinesin family member 4A (KIF4A), another chromokinesin. Cells that underwent chromosome missegregation took relatively longer time to align chromosomes in both control and Kid/KIF4A-depleted cells. Tracking of late-aligning chromosomes showed that they exhibit a higher rate of lagging chromosomes. Intriguingly, the metaphase of cells that underwent chromosome missegregation was shortened, and delaying anaphase onset ameliorated the increased chromosome missegregation. These data suggest that late-aligning chromosomes do not have sufficient time to establish bi-orientation, leading to chromosome missegregation. Our data imply that delayed chromosome alignment is not only a consequence, but also a cause of defective bi-orientation establishment, which can lead to chromosomal instability in cells without severe mitotic defects.

## 1. Introduction

For proper mitotic chromosome segregation, sister kinetochores on a replicated chromosome pair have to attach to microtubules from opposite spindle poles, which is called amphitelic attachment, or bi-orientation. As kinetochores can attach to microtubules only after nuclear envelope breakdown (NEBD) in cells that undergo open mitosis, elaborate mechanisms exist to ensure the establishment of bi-orientation for all the sister kinetochores at every mitosis. Failure in this process leads to chromosome missegregation that results in abnormal numbers of chromosomes, known as aneuploidy, which is common in cancer [1]. Chromosomal instability (CIN), a condition in which chromosome missegregation occurs at a high rate, usually underlies the occurrence of aneuploidy in cancer [2,3]. During the process to establish bi-orientation, erroneous kinetochore attachments to microtubules can occur, such as monotelic, syntelic, or merotelic attachment [4,5]. Aurora B, a mitotic kinase, plays a role in the correction of these erroneous attachments by destabilizing them, while the spindle assembly checkpoint (SAC) halts mitotic progression in the presence of unattached kinetochores, allowing time for error correction. Among these erroneous attachments, merotelic attachment, in which a single kinetochore attaches to microtubules from both spindle poles, is considered a major cause of CIN, as it is not sensed by the SAC because the kinetochore is attached to microtubules and under tension exerted by pulling force from both spindle poles that stabilizes the attachment [6]. Uncorrected merotelic attachment gives rise to lagging chromosomes, which can be a cause of both numerical and structural chromosome abnormalities as well as formation of micronuclei [2]. Several conditions that promote the formation of merotelic attachment have been reported, including overamplification of centrosomes, weakening of Aurora B activity, and hyperstabilization of microtubules [2]. However, the whole picture of the causes for merotelic attachment that leads to CIN in cancer has not been revealed.

During prometaphase, chromosomes move to the spindle equator, referred to as congression. Chromosome congression is carried out by two different mechanisms depending on the position of chromosomes relative to spindle poles at NEBD [7,8,9]. When chromosomes are between spindle poles at NEBD and directly incorporated in the spindle, bi-orientation is quickly established. Bi-oriented kinetochores attach to the ends of bundled microtubules (K-fiber), which is called end-on attachment, and chromosome congression occurs through elongation or shrinkage of K-fibers. In contrast, a fraction of chromosomes that are located outside of the spindle attach to the lateral surface of microtubules via kinetochores, called lateral attachment. Laterally-attached chromosomes move along microtubules initially towards spindle poles in a Dynein-dependent manner, then to the spindle equator that is driven by chromokinesins and centromere protein E (CENP-E), achieving congression. Bi-orientation is established at the spindle equator, through conversion of lateral attachment to end-on attachment. We and others have reported that cells defective in both lateral and end-on attachment exhibit an extensive loss of connection between chromosomes and microtubules, underscoring the role of lateral attachment in incorporating chromosomes into the spindle [9,10]. In line with previous reports [11,12], we also uncovered the differential contribution of a chromokinesin Kid and CENP-E to the congression of laterally-attached chromosomes, which was clarified under the condition where end-on attachment is suppressed [13]. However, whether congression of chromosomes through lateral attachment contributes to mitotic fidelity in physiological condition has not been revealed yet. A plausible advantage for laterally-attached chromosomes to establish bi-orientation at the spindle equator is that microtubules from each spindle pole exist at a similar density, allowing efficient bi-orientation formation (Figure 1i) [14]. In contrast, when chromosomes are close to one of the spindle poles, kinetochores are prone to attach to microtubules from the closer spindle pole that exist at a higher density, resulting in the formation of erroneous attachments (Figure 1ii). To prevent this, Aurora A, which is concentrated at spindle poles, destabilizes kinetochore-microtubule attachments around spindle poles [15,16]. Therefore, we hypothesized that when chromosome alignment is delayed, late-aligning chromosomes staying near a spindle pole for a prolonged period are difficult to establish bi-orientation, resulting in the increase in the rate of chromosome missegregation. We examined this hypothesis in cancer cell lines depleted of chromokinesins, Kid and KIF4A [8]. We found that late-aligning chromosomes exhibit an increased chromosome missegregation rate, which is supposed to result from a shortened metaphase that is insufficient to establish bi-orientation. Delay or failure in chromosome alignment has been recognized as a consequence of problems in kinetochore-microtubule attachments, but our findings suggest that delayed chromosome alignment per se can be a cause of erroneous kinetochore-microtubule attachments.

## 2. Materials and Methods

### 2.1. Antibodies

Monoclonal mouse antibody against α-tubulin B-5-1-2 (T5168, Merck, Darmstadt, Germany) was used for immunofluorescence analysis (IF) and Western blotting (WB) at 1:2000. Polyclonal rabbit antibodies against the following proteins were used: anti-Kid (AKIN12, Cytoskeleton Inc., Denver, CO, USA) for WB at 1:2000; anti-KIF4A (A301-074A, BETHYL, Montgomery, TX, USA) for WB at 1:2000, and anti-CENP-A (#2186, Cell Signaling Technology, Danvers, MA, USA) for IF at 1:400. Human auto-antiserum against centromeric antigen (ACA; IF 1:30,000) was a gift from T. Hirota.

### 2.2. RNA Interference

RNA oligonucleotides targeting human Kid (#1) and KIF4A were purchased from J-BioS (Asaka, Japan). The sequences were 5′-AAGAUUGGAGCUACUCGUCGUTT-3′ [17] and 5′-GAAAGAUCCUGGCUCAAGATT-3′ [18], respectively. The sequence of another RNA oligonucleotide targeting Kid (#2) was 5′-GGGACCUGUUAAGCUGUCUCAGAAA-3′ (Stealth, Thermo Fisher Scientific, Waltham, MA, USA). For control small interfering RNA (siRNA), Stealth RNAi^TM^ siRNA Negative Control Med GC duplex #2 was used (Thermo Fisher Scientific). RNA duplexes (50 nM) were transfected into cells using Lipofectamine RNAiMAX reagent (Thermo Fisher Scientific).

### 2.3. Cell Culture and Synchronization

HCT116, RPE (retinal pigment epithelium)-1 and HeLa Kyoto (HeLa) cells were grown at 37 °C in a 5% CO_2_ atmosphere in Dulbecco’s Modified Eagle Medium (DMEM; Nacalai Tesque, Kyoto, Japan), supplemented with 10% fetal bovine serum (FBS; Thermo Fisher Scientific). For RNA interference (RNAi) experiments, cells were transfected with siRNAs for 12 h and then synchronized by thymidine treatment (2 mM) for 24 h. Ten hours after release from the thymidine block, synchronized mitotic cells were used for further analysis. In Figure 2B,C and Figure 3B,C, cells were treated with Z-Leu-Leu-Leu-CHO (MG132; 10 mM stock solution in dimethyl sulfoxide (DMSO) was diluted to a final concentration of 20 μM) (Merck) for 1 h before fixation. In Figure 6B, cells were treated with proTAME (20 mM stock solution in DMSO was diluted to a final concentration of 3 μM) (Boston Biochem, Cambridge, MA, USA) eight hours after release from the thymidine block, then subjected to live cell imaging. In Figure 6C,D, cells were treated with proTAME (3 μM) for 2 h or MG132 (20 μM) for 1 h. In case of MG132 treatment, cells were used for experiments 2 h after release from the drug. In Figure 6B–D, the same amount of DMSO was added to control samples without drug treatments.

### 2.4. Immunofluorescence Analysis

Cells were grown on a glass coverslip and fixed with methanol at −20 °C for 5 min. Before fixation, cells were pre-extracted with PHEM buffer (60 mM piperazine-1,4-bis(2-ethanesulfonic acid) (PIPES), 25 mM 4-(2-hydroxyethyl)-1-piperazineethanesulfonic acid) (HEPES), 10 mM ethylene glycol tetraacetic acid (EGTA), 2 mM MgCl_2_, pH 7.4) containing 0.1% Triton X-100 for 1 min. Fixed cells were treated with 3% bovine serum albumin (BSA; Fraction V) (Wako, Osaka, Japan) for 30 min at room temperature and then incubated with primary antibodies for overnight at 4 °C, followed by incubation with secondary antibodies coupled with Alexa-Fluor-488/568 at 1:2000 (Thermo Fisher Scientific) for 1 h at room temperature. Antibodies were incubated in phosphate buffered saline (PBS; 137 mM NaCl, 2.7 mM KCl, 10 mM Na_2_HPO_4_, and 1.8 mM KH_2_PO_4_, pH 7.4) supplemented with 0.01% Triton X-100 and 1% BSA. To stain DNA, 4′,6-diamidino-2-phenylindole (DAPI) was added at 1 μg/mL. After final washes, cells were mounted with Dako fluorescence mounting medium (Agilent, Santa Clara, CA, USA). Z-image stacks were captured in 0.2 μm increments on an IX-71 inverted microscope (Olympus, Tokyo, Japan) controlled by DeltaVision softWoRx (version 7.0.0., GE Healthcare, Chicago, IL, USA) using a 100 × 1.40 NA Plan Apochromat oil objective lens (Olympus). Deconvolution was performed when necessary. Image stacks were saved as TIFF files and represented as maximum intensity projections. In Figure 2B,F,H, Figure 3H, and Appendix A, ten to fifty z-stack images were projected. In Figure 3B,F, z-image stacks were captured in 1 μm increments, and fifteen sections were projected.

### 2.5. Western Blotting

Forty-eight hours after siRNA transfection, asynchronous cells were lysed in TEN-N buffer (1% NP-40, 100 mM NaCl, 10 mM Tris-HCl, pH 7.5, and 1 mM EDTA). Protein concentration was measured by the Bio-Lad Protein assay kit (Bio-Rad, Hercules, CA, USA). Extracted proteins were boiled for 10 min with 4 × NuPAGE® LDS (lithium dodecyl sulfate) sample buffer (Thermo Fisher Scientific). Proteins were separated by NuPAGE® SDS (sodium dodecyl sulfate) PAGE (polyacrylamide gel electrophoresis)-Gel System (Thermo Fisher Scientific), electroblotted onto a polyvinylidene difluoride membrane (Amersham Hybond-P, GE Healthcare), and reacted with appropriate primary antibodies overnight at 4 °C for immunodetection. Blocking and antibody incubation were performed in 3% non-fat dry milk. Proteins were detected using horseradish peroxidase-labeled secondary antibodies (1:3000, Santa Cruz Biotechnology, Dallas, TX, USA) for 1 h at room temperature and enhanced chemiluminescence, according to the manufacturer’s instructions (GE Healthcare).

### 2.6. Live Cell Imaging

HCT116 cells and HeLa Kyoto cells expressing EGFP (enhanced green fluorescent protein)-α-tubulin, EGFP-CENP-A, and H2B-mCherry were grown in glass chambers (Thermo Fisher Scientific). Thirty minutes before imaging, the medium was changed to pre-warmed Leibovitz’s L-15 medium (Thermo Fisher Scientific) supplemented with 20% FBS and 20 mM HEPES, pH 7.0. Recording was performed in a temperature-controlled incubator at 37 °C. Z-series of five mCherry image sections in 3 μm increments were captured every 2 min. Images were obtained using an IX-71 inverted microscope (Olympus) controlled by DeltaVision softWoRx (GE Healthcare) using a 20 × 0.75 NA UPlanS Apochromat objective lens (Olympus). Deconvolution was performed when necessary, and image stacks were projected.

### 2.7. Statistical Analysis

A Mann-Whitney *U* test was used for comparison of dispersion, and a two-sided Student’s *t*-test was used for the comparison of the average. Dispersibility of each category was validated by a two-sided *F*-test before Student’s *t*-test. A two-sided *F*-test was also used to analyze the results of chromosome spreads. If the result of *F*-test was an unequal variance, statistical significance between samples was validated by a two-sided Welch’s *t*-test. A Chi-squared test was used for comparison between measured value and theoretical value. However, when more than 20% of cells in a table have expected frequencies <5, Fisher’s exact test was used instead of Chi-squared test. R (version 3.2.2, R Core Team (2015). R: A language and environment for statistical computing. R Foundation for Statistical Computing, Vienna, Austria. URL https://www.R-project.org/.) and Microsoft Excel were used to perform statistical analysis.

### 2.8. Chromosome Spread

Cells were grown on 10 cm plates and transfected with siRNAs for 48 h. Cells were treated with nocodazole (2 μM) for 6 h. Mitotic cells were harvested by shake off and hypotonically swollen in 20% culture medium with 80% tap water for 5 min. Cells were fixed with Carnoy’s solution (methanol: acetic acid 3:1) and then dropped onto slide glasses, which were pre-wetted with 70% ethanol. Slides were dried using flame and stained with DAPI. Images were obtained using IX-83 inverted microscope (Olympus) controlled by cellSens (Olympus) using a 100× 1.40 NA UPlanS Apochromat oil objective lens (Olympus).

### 2.9. Photoactivation

An expression vector for H2B-PA (photoactivatable)-GFP (green fluorescent protein) was constructed by inserting coding sequences for histone H2B and PA-GFP into the pIREShyg3 vector (Takara-bio, Kusatsu, Japan). HeLa Kyoto cells stably expressing H2B-PA-GFP were grown in glass bottom dishes (MatTek, Ashland, MA, USA). In Figure 5, cells were treated with a CENP-E inhibitor, GSK-923295 (200 nM) (Cayman Chemical, Ann Arbor, MI, USA), for 1 h and washed out before live cell imaging. Thirty minutes before imaging, the medium was changed to pre-warmed Leibovitz’s L-15 medium supplemented with 20% FBS and 20 mM HEPES, pH 7.0. Aligned chromosomes in the metaphase plate, or late-aligning chromosomes in the vicinity of a spindle pole were photoactivated using 1 pulse of <1000 ms duration with 1% 405 nm laser power. Recording was performed in a temperature-controlled incubator at 37 °C. Z-series of five to ten sections in 2 μm increments were captured every 3 min in Figure 5C,D, or every 1 min in Appendix A. Images were obtained using the TCS SP8 confocal microscope (Leica, Wetzlar, Germany) controlled by LAS X (Leica) using a HC PL APO 63 × /1.40 Oil objective lens (Leica). Median filter (ImageJ, Rasband, W.S., ImageJ, U. S. National Institutes of Health, Bethesda, MD, USA, https://imagej.nih.gov/ij/, 1997–2018) was used for denoising in Figure 5C.

## 3. Results

### 3.1. Cells Depleted of Kid and KIF4A Showed Delayed Chromosome Alignment and Increased Chromosome Missegregation

We have recently shown that two motor proteins, Kid and CENP-E, differentially contribute to chromosome congression in human cells [13]. CENP-E is a microtubule plus end-directed motor of the kinesin-7 family that localizes to kinetochores and transport laterally-attached chromosomes towards the spindle equator preferentially on stabilized microtubules [13,19,20,21]. On the other hand, Kid, a chromokinesin of the kinesin-10 family, localizes to chromosome arms and moves chromosomes away from spindle poles [22,23,24]. In contrast to depletion of CENP-E, which results in misalignment of several chromosomes near spindle poles (polar chromosomes) [20], when Kid was depleted in HeLa cells, an aneuploid cell line derived from cervical cancer, chromosomes were properly aligned at the spindle equator [13]. However, there was a small but significant increase in the time required to align all the chromosomes, showing that chromosome alignment was delayed (Figure 1e in reference [13]). We studied whether this delay in chromosome alignment leads to increased chromosome missegregation in anaphase and telophase cells (Appendix A). We found increased incidence of chromosome missegregation in Kid-depleted cells compared to mock-treated cells (Appendix A). Additionally, we observed the effect of Kid depletion in RPE-1 cells, a normal cell-derived cell line. The rate of chromosome missegregation in RPE-1 cells was very low, and we could not detect a significant increase in chromosome missegregation by Kid depletion (mock: 0.29% (2/688), Kid si: 0.72% (5/698), *p* = 0.264, chi-squared test). However, when we measured the distribution of chromosome number in chromosome spreads, the percentage of cells with a modal number of chromosomes (n = 46) decreased in Kid-depleted cells, while cells showing aneuploidy increased (Appendix AC). These data suggest the link between delayed chromosome alignment and increase in the rate of chromosome missegregation in Kid-depleted cells.

To corroborate the result, we observed HCT116 cells, which is a chromosomally stable cell line derived from colorectal cancer, depleted of Kid (Figure 2A). As seen in HeLa cells, chromosome alignment occurred properly in HCT116 cells depleted of Kid with two independent siRNAs (Figure 2B,C), determined in fixed cell samples after treatment with MG132, a proteasome inhibitor that arrests cells in metaphase, to discriminate sustained chromosome misalignment from transient chromosome misalignment. However, in a live imaging of cells expressing histone H2B-mCherry, the time required for the alignment was slightly but significantly increased (Figure 2D,E). Then, we examined chromosome missegregation, and found that cells depleted of Kid with two independent siRNAs exhibited an increased frequency of lagging chromosomes (Figure 2F,G). Moreover, we quantified interphase cells containing micronuclei (Figure 2H), which formed when lagging chromosomes failed to join other chromosomes in telophase [6]. We found a significant increase of cells with micronuclei in Kid-depleted cells (Figure 2I), confirming the increased chromosome missegregation in these cells. Next, we counted the chromosome number in chromosome spreads, and found that the percentage of cells with modal chromosome number (n = 45) decreased, while cells with abnormal chromosome numbers increased (Appendix A). These data confirmed the increased chromosome missegregation in Kid-depleted cells, which was accompanied with delayed chromosome alignment.

Additionally, we addressed the effect of depletion of KIF4A, another chromokinesin of the kinesin-4 family, which was also reported to be involved in chromosome congression [12,24] (Figure 3A). KIF4A-depleted cells did not show an increase in chromosome misalignment (Figure 3B,C), however, the time required for chromosome alignment was increased slightly but significantly (Figure 3D,E), as in Kid-depleted cells. KIF4A-depleted cells also showed an increase in the appearance of lagging chromosomes (Figure 3F,G), as well as the rate of micronuclei-containing cells (Figure 3H,I) and the percentage of cells with abnormal chromosome numbers (Appendix A).

Collectively, our data suggest that depletion of chromokinesins involved in chromosome congression delays chromosome alignment and increases the rate of chromosome missegregation.

### 3.2. Cells That Underwent Chromosome Missegregation Exhibit Elongated Prometaphase and Shortened Metaphase

To verify the relationship between delayed chromosome alignment and increased chromosome missegregation, we observed mitosis in cells with or without Kid depletion, and compared the duration of prometaphase and metaphase depending on the presence of chromosome segregation errors. As shown in Appendix AA, the duration of prometaphase in Kid-depleted cells was longer than that in mock-treated cells, as already shown, while the metaphase was shortened. Confirming the previous result, Kid-depleted cells showed a higher rate of chromosome missegregation than mock-treated cells, and cells that exhibited chromosome missegregation spent a longer time in prometaphase in both mock and Kid-depleted cells (Figure 4A), showing the relationship between delayed chromosome alignment and increased chromosome missegregation among the same population of cells. On the other hand, duration of metaphase was shorter for mock-treated cells that underwent chromosome missegregation (Figure 4A). Elongated prometaphase and shortened metaphase for cells that underwent chromosome missegregation was also seen in HeLa cells (Appendix AB), supporting the finding in HCT116 cells. Metaphase length in Kid-depleted cells did not differ depending on the occurrence of chromosome missegregation (Figure 4A). However, when we quantified the ratio of metaphase length to the total length of prometaphase and metaphase in each cell, it was smaller in cells that underwent chromosome missegregation not only for mock-treated cells (0.34 ± 0.17 vs. 0.59 ± 0.12), but also for Kid-depleted cells (0.30 ± 0.21 vs. 0.41 ± 0.15) (Figure 4B). The ratio was bigger in mock-treated cells compared to Kid-depleted cells for cells that exhibited proper chromosome segregation (0.59 ± 0.12 vs. 0.41 ± 0.15), but comparable for cells that underwent chromosome missegregation (0.34 ± 0.17 vs. 0.30 ± 0.21) (Figure 4B). KIF4A-depleted cells also showed a longer prometaphase and a shorter metaphase for cells that underwent chromosome missegregation (Appendix AC). These data indicate that cells that underwent chromosome missegregation preferentially show a delayed chromosome alignment, while metaphase in these cells is accordingly shortened, which is seen not only in cells depleted of Kid or KIF4A, but also in mock-treated cells.

### 3.3. Late-Aligning Chromosomes Have a Higher Chance to Undergo Chromosome Missegregation

To unravel the relationship between delayed chromosome alignment and increased chromosome missegregation, we developed a method to track chromosomes through prometaphase to anaphase. We established a HeLa cell line expressing histone H2B tagged with photoactivatable (PA)-GFP, in which global chromosome alignment can be followed by basal expression of the GFP signal (Figure 5A). We selected cells in which a majority of chromosomes aligned at the spindle equator, and photoactivated late-aligning chromosomes, followed by observation of chromosome segregation in live cell imaging (Figure 5A). As a control, chromosomes aligned at the spindle equator were photoactivated and tracked (Figure 5A). First, we observed cells without drug treatment. Photoactivated chromosomes are discriminated by brighter signals compared to basal GFP signals on other chromosomes (Appendix A), which enabled us to follow their fates. As shown in Appendix AB, late-aligning chromosomes exhibited a higher incidence of lagging chromosomes compared to aligned chromosomes, although the experiment was time-consuming and we were unable to collect a sufficient number of samples to show statistical significance. To overcome this problem, we treated cells with a CENP-E inhibitor, which increases cells containing several polar chromosomes (Figure 5B). Soon after washing out the drug, we photoactivated late-aligning or aligned chromosomes in several cells consecutively at a time, and tracked the photoactivated chromosomes (Figure 5C). We found a significant increase in the frequency of lagging chromosomes for late-aligning chromosomes (Figure 5D), confirming the trend of cells without drug treatment. As multiple chromosomes are inevitably photoactivated for aligned chromosomes, in which precise numbers were difficult to determine, the difference in the chromosome missegregation rate between late-aligning and aligned chromosomes may be underestimated. These data suggest that late-aligning chromosomes, which are responsible for delayed chromosome alignment, contribute to the increase in the rate of lagging chromosomes, probably due to the formation of erroneous kinetochore-microtubule attachments.

### 3.4. Delaying Anaphase Onset Attenuated Chromosome Missegregation Caused by Delayed Chromosome Alignment

It was reported that correction of erroneous kinetochore-microtubule attachments occurs not only in prometaphase, but also in metaphase [14]. Our data that cells that underwent chromosome missegregation exhibit elongated prometaphase and shortened metaphase (Figure 4) suggest that these cells do not have sufficient time to correct erroneous kinetochore-microtubule attachments during metaphase. Therefore, we examined whether increasing the metaphase duration by delaying anaphase onset improved the mitotic fidelity of cells showing delayed chromosome alignment. To delay anaphase onset, we treated cells with a low dose of proTAME, an inhibitor of anaphase promoting complex/cyclosome (APC/C) (Figure 6A). We confirmed that HCT116 cells treated with 3 μM proTAME did not arrest cells in metaphase, but delayed anaphase onset, while prometaphase duration did not change (Figure 6B). As another means to delay anaphase onset, we transiently treated cells with MG132 that inhibit anaphase onset (Figure 6A). To delay chromosome alignment, we depleted Kid in HCT116 cells, then treated cells with or without proTAME/MG132, and observed chromosome missegregation or micronuclei formation (Figure 6C,D). As shown previously, Kid-depleted cells showed an increased rate of lagging chromosomes as well as micronuclei formation (Figure 6C,D). Intriguingly, both proTAME and MG132 treatment significantly reduced the frequency of lagging chromosomes and micronuclei formation in Kid-depleted cells (Figure 6C,D). Our data suggest that lack of time for error correction during metaphase promotes missegregation of late-aligning chromosomes, which can be rescued by delaying anaphase onset.

## 4. Discussion

In this paper, we addressed whether delayed chromosome alignment compromises mitotic fidelity, under the assumption that late-aligning chromosomes have a difficulty in establishing bi-orientation (Figure 7). When chromosome alignment is delayed, a fraction of chromosomes stay longer near spindle poles. Kinetochore-microtubule attachments in such late-aligning chromosomes are unstable due to Aurora A-dependent destabilization, or would be erroneous because of unequal distribution of microtubules from each spindle pole. Even though these chromosomes finally align to the spindle equator, the metaphase is shortened in association with an elongated prometaphase, which is insufficient to correct erroneous attachments and establish bi-orientation. As a result, the remaining erroneous attachments lead to the formation of lagging chromosomes in anaphase, and a fraction of lagging chromosomes cause aneuploidy or micronuclei formation.

To avoid a secondary effect of problems in the formation of end-on attachment, we exploited the depletion of a chromokinesin Kid throughout the study to delay chromosome alignment, which localizes to chromosome arms and specifically functions in chromosome congression [8]. Although chromosomes aligned properly in Kid-depleted cells, they showed delayed chromosome alignment and increased chromosome missegregation in two different cell lines, HeLa and HCT116 cells, supporting our assumption. In contrast, RPE-1 cells did not show a significant increase in chromosome missegregation after Kid depletion, although aneuploid cells were increased (Appendix AC). Intrinsic difference in mitotic fidelity between normal cells and cancer cells, the nature of which is poorly understood, may underlie the different responses to Kid depletion. Rescue experiments to express RNAi-resistant Kid in Kid-depleted cells were unsuccessful, because Kid overexpression showed chromosome misalignment due to defective spindle organization (data not shown) [13]. However, two different siRNAs showed similar results (Figure 2), excluding the possibility of off-target effects. Depletion of KIF4A, another chromokinesin working in chromosome congression [12,24], also showed delayed chromosome alignment and increased chromosome missegregation (Figure 3), corroborating our hypothesis. However, it is important to note that KIF4A has other functions in mitosis, such as chromosome condensation [18,25] and spindle formation, which may also be related to mitotic fidelity [8,26].

Chromosome alignment is physiologically delayed for several reasons. First, the position of chromosomes relative to the spindle in the early prometaphase is crucial for efficient chromosome capture by microtubules and chromosome alignment to the spindle equator. In contrast to chromosomes positioned inside the spindle, which instantly achieve bi-orientation and align at the spindle equator, chromosomes outside the spindle are captured by microtubules at their lateral surface, and transported towards spindle poles in a Dynein-dependent manner before chromosome congression along microtubules, which can cause delayed chromosome alignment [7,8,27]. The position of the spindle poles at NEBD also influences the efficiency of spindle assembly and chromosome alignment. When spindle poles are separated at the opposite side of the nucleus at NEBD, a majority of chromosomes are locating between spindle poles and quickly incorporated in the spindle by forming bi-orientation, referred to as “prophase pathway”. In contrast, when spindle poles are closer to each other and at the same side of the nucleus at NEBD, chromosomes align at the surface of the nascent spindle via lateral attachment (prometaphase rosette), and establish bi-orientation at the spindle equator after congression, which is called the “prometaphase pathway” [9,28,29,30]. It was reported that prophase pathway cells achieve spindle formation earlier and have higher mitotic fidelity than prometaphase pathway cells, underscoring the relationship between elongated prometaphase and increased chromosome missegregation [30]. In line with this notion, we found that prometaphase duration was longer in both mock- and Kid/KIF4A-depleted cells that underwent chromosome missegregation (Figure 4A and Appendix A).

In pathological conditions, it is well known that the depletion of molecules directly involved in end-on attachment show chromosome misalignment and prometaphase arrest in a SAC-dependent manner. On the other hand, depletion of molecules indirectly involved in the formation of end-on attachment and bi-orientation, such as microtubule plus-end tracking proteins and motor proteins, often show delayed chromosome alignment even when chromosome misalignment is mild, and exhibit increased chromosome missegregation [8,26]. In these cases, delayed chromosome alignment is primarily a result of problems in kinetochore-microtubule attachment, however, it may also exacerbate the increase in chromosome missegregation.

To clarify the causal relationship between delayed chromosome alignment and increased chromosome missegregation, we developed an assay to track late-aligning chromosomes, and found that they exhibit a higher incidence of lagging chromosomes (Figure 5C,D). This data suggest that late-aligning chromosomes are inclined to form erroneous kinetochore-microtubule attachments. To confirm the tendency found in normal cycling cells (Appendix A), we treated cells with a CENP-E inhibitor to enrich cells containing polar chromosomes. This is an artificial condition that is supposedly different from spontaneous alignment delay, but still represents chromosomes locating in the vicinity of spindle poles, in which kinetochores are mostly devoid of microtubules [12,31,32]. We need to bear in mind, however, that CNEP-E has been suggested to play a role in stabilization of end-on attachment [31,33]. Therefore, there remains a possibility that a residual effect of the CENP-E inhibitor after washout might affect bi-orientation establishment. Even though most of the lagging chromosomes we found in the tracking finally segregated equally as previously reported [34], a small fraction of lagging chromosomes may lead to aneuploidy or micronuclei formation. We compared chromosome missegregation rate between late-aligning and aligned chromosomes; however, an ideal comparison would be the tracking of chromosomes randomly-labeled at the beginning of mitosis, which was difficult because multiple chromosomes are inevitably labeled by photoactivation of clustered chromosomes in early mitosis. Differential painting of chromosomes, which is now feasible using CRISPR/Cas9 technology [35], may circumvent the problem.

There are several mechanisms to prevent the formation and stabilization of end-on attachment on chromosomes near spindle poles. Aurora A, which mainly localizes to spindle poles, destabilizes end-on attachment of chromosomes near spindle poles [15,16], and Dynein-dependent minus-end directed chromosome transport facilitates destabilization of end-on attachment against chromokinesin-mediated polar ejection force [12]. Chromosome congression via lateral attachment may also contribute to prevent premature formation of end-on attachment near spindle poles. In spite of these mechanisms, end-on attachment can be formed near spindle poles in some contexts, as exemplified by the situation when mono-oriented (end-on attachment is formed in one of the sister kinetochores) chromosomes near a spindle pole move along microtubules via lateral attachment of another sister kinetochore in a CENP-E dependent manner [20]. Therefore, erroneous kinetochore-microtubule attachments on late-aligning chromosomes may be formed not only at the spindle equator, but also near spindle poles, although we were not able to discriminate merotelically-attached microtubules on late-aligning chromosomes near spindle poles in immunofluorescence staining among bundles of spindle microtubules (data not shown).

We showed that cells that underwent chromosome missegregation exhibit a shortened metaphase in addition to an elongated prometaphase (Figure 4). It is known that the SAC response scales with the number of unattached chromosomes [36], and merotelic attachment is not sensed by the SAC [6]. Therefore, it is understandable that the metaphase becomes shortened when the alignment of a small number of chromosomes is delayed, which may be insufficient to establish bi-orientation for some of the late-aligned chromosomes. The situation would be different when the alignment of many chromosomes is delayed. Even when late-aligning chromosomes maintain lateral attachment when they reach the spindle equator, a lack of time for the correction of errors accrued during conversion to end-on attachment may lead to increased chromosome missegregation. Intriguingly, reduced mitotic fidelity in Kid-depleted cells was ameliorated by delaying the anaphase onset (Figure 6C,D), supporting the notion that a shortened metaphase is responsible for the increased chromosome missegregation. We used MG132 treatment not only to discriminate permanent chromosome misalignment from transient chromosome misalignment (Figure 2B,C), but also to delay the anaphase onset (Figure 6C,D), which would allow these transiently-misaligned chromosomes a sufficient time to establish bi-orientation. Our data are in agreement with recent reports that delaying anaphase onset suppresses chromosome missegregation in cancer [37,38]. The reason why prometaphase was shortened in Kid-depleted cells by proTAME treatment (Figure 6B) is currently unknown.

In conclusion, our data support the hypothesis that delayed chromosome alignment increases the rate of chromosome missegregation. This hypothesis can account for a cause of spontaneous chromosome missegregation in cancer cells, which can be a potential cause of CIN. It has now been recognized that chromosome missegregation rates in cancer cells are kept within a range to balance genomic heterogeneity and cellular fitness [39]. A mild increase of chromosome missegregation caused by delayed chromosome alignment may trigger clonal evolution that leads to cancer formation and progression. Concerning anti-cancer strategy, increasing the level of CIN over the threshold tolerable for cancer cells has been postulated to selectively eradicate cancer cells [2]. In this respect, impairing chromosome alignment can be a way to increase the level of CIN to kill cancer cells while minimizing the adverse effect to normal cells.

## Figures and Tables

**Figure 1 biomolecules-09-00010-f001:**
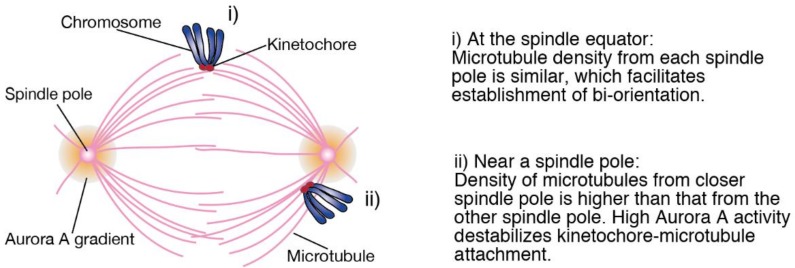
A schematic diagram showing the efficiency of bi-orientation establishment depending on the position of chromosomes on the spindle. Chromosomes that attach to spindle microtubules laterally at the spindle equator (**i**) and near a spindle pole (**ii**) are shown.

**Figure 2 biomolecules-09-00010-f002:**
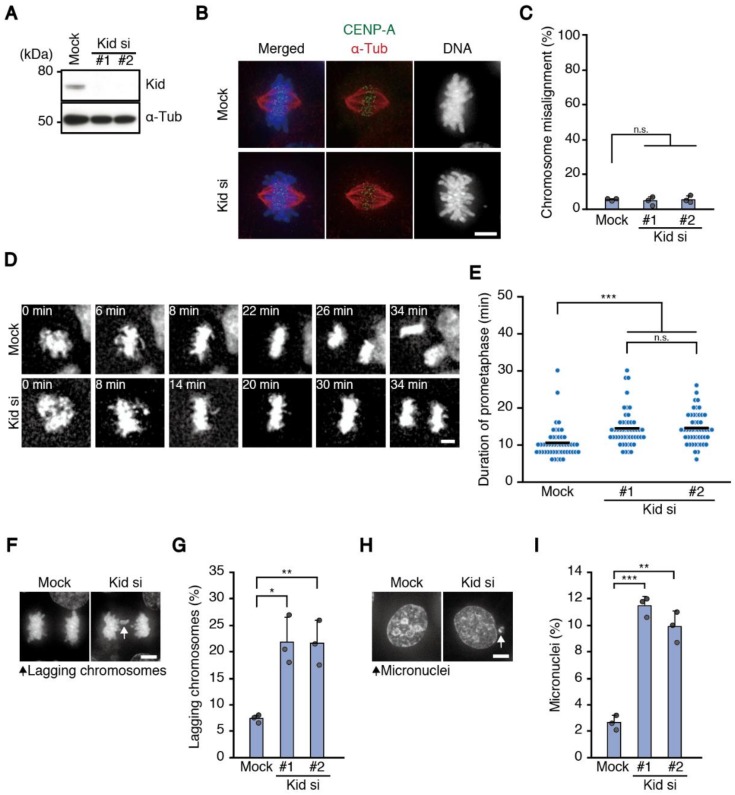
Cells depleted of Kid show delayed chromosome alignment and increased chromosome missegregation. (**A**) Efficiency of RNAi for Kid in HCT116 cells. Lysate of cells transfected with siRNAs against Kid was subjected to immunoblot analysis using antibodies as indicated; (**B**) chromosome alignment in cells depleted of Kid. HCT116 cells were transfected with the siRNAs for Kid shown in **A**, then fixed after 1 h treatment with MG132 and immunostained with an antibody against centromere protein A (CENP-A; green) and α-tubulin (red). DNA was stained with 4′,6-diamidino-2-phenylindole (DAPI; blue). Only a cell depleted of Kid with one of the siRNAs (#1) is shown. Scale bar: 5 μm; (**C**) proportion of cells with misaligned chromosomes. HCT116 cells treated as in (**B**) were observed. For each condition, 100 cells were observed. Error bars represent standard deviation (SD) of three independent experiments, and the average of each experimental result is shown as a dot. n.s., not statistically significant (Student’s *t*-test); (**D**) mitotic progression of mock and Kid-depleted cells. HCT116 cells expressing histone H2B-mCherry were transfected with or without the siRNAs for Kid and subjected to live cell imaging. Time from nuclear envelope breakdown is shown. Scale bar: 5 μm; (**E**) duration of prometaphase. For cells treated as in (**D**), time from nuclear envelope breakdown to completion of chromosome alignment was measured in 50 mitotic cells for each condition. The average is indicated with a bar. n.s., not statistically significant, *** *p* < 0.0005 (Mann-Whitney *U* test); (**F**) chromosome missegregation in cells depleted of Kid. HCT116 cells were transfected with the siRNAs for Kid. After fixation, DNA was stained with DAPI, then, anaphase and telophase cells were observed. Only a cell depleted of Kid with one of the siRNAs (#1) is shown. An arrow indicates lagging chromosomes. Scale bar: 5 μm; (**G**) proportion of cells with lagging chromosomes. For each condition, 200 HCT116 cells treated as in (**F**) were observed. Error bars represent SD of three independent experiments, and the average of each experimental result is shown as a dot. * *p* < 0.05, ** *p* < 0.005 (Student’s *t*-test); (**H**) micronuclei formation in cells depleted of Kid. HCT116 cells were treated as in (**F**) and interphase cells were observed. Only a cell depleted of Kid with one of the siRNAs (#1) is shown. An arrow indicates a micronucleus. Scale bar: 5 μm; (**I**) proportion of cells with micronuclei. HCTT16 cells treated as in (**F**) were observed for the presence of micronuclei as shown in (**H**). For each condition, 1000 cells were observed. Error bars represent SD of three independent experiments, and the average of each experimental result is shown as a dot. ** *p* < 0.005, *** *p* < 0.0005 (Student’s *t*-test).

**Figure 3 biomolecules-09-00010-f003:**
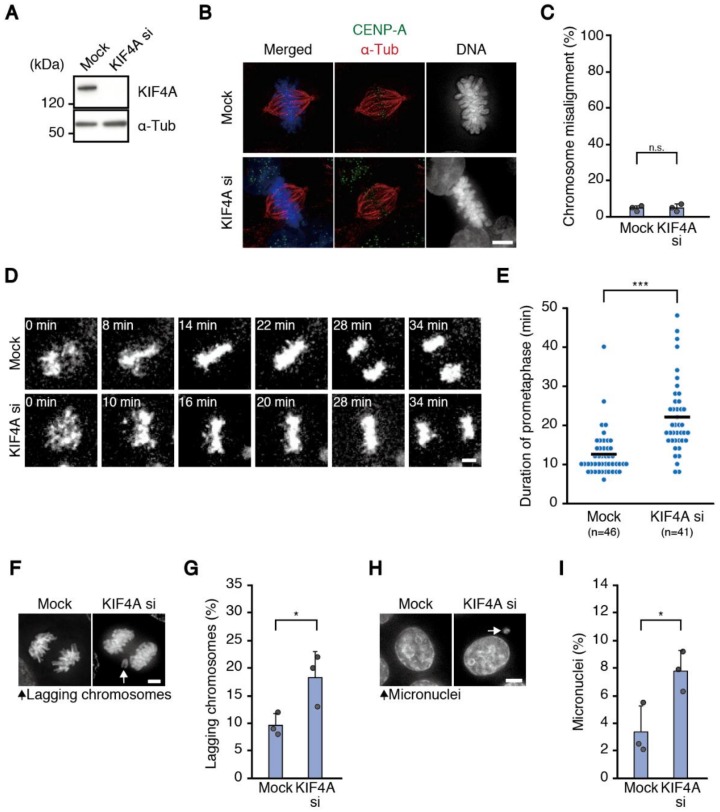
Cells depleted of KIF4A show delayed chromosome alignment and increased chromosome missegregation. (**A**) Efficiency of RNAi for KIF4A in HCT116 cells. Lysate of cells transfected with an siRNA against KIF4A was subjected to immunoblot analysis using antibodies as indicated; (**B**) chromosome alignment in cells depleted of KIF4A. HCT116 cells were transfected with the siRNA against KIF4A shown in (**A**), fixed after 1 h treatment with MG132 and immunostained with an antibody against CENP-A (green) and α-tubulin (red). DNA was stained with DAPI (blue). Scale bar: 5 μm; (**C**) proportion of cells with misaligned chromosomes. HCT116 cells treated as in (**B**) were observed. For each condition, 100 cells were observed. Error bars represent SD of three independent experiments, and the average of each experimental result is shown as a dot. n.s., not statistically significant (Student’s *t*-test); (**D**) mitotic progression of mock and KIF4A-depleted cells. HCT116 cells expressing H2B-mCherry were transfected with or without the siRNA for KIF4A and subjected to live cell imaging. Time from nuclear envelope breakdown is shown. Scale bar: 5 μm; (**E**) duration of prometaphase. For cells treated as in (**D**), time from nuclear envelope breakdown to completion of chromosome alignment was measured for each condition. Number of cells observed is shown. The average is indicated with a bar. *** *p* < 0.0005 (Mann-Whitney *U*-test); (**F**) chromosome missegregation in cells depleted of KIF4A. HCT116 cells were transfected with the siRNA for KIF4A. After fixation, DNA was stained with DAPI, then, anaphase and telophase cells were observed. An arrow indicates lagging chromosomes. Scale bar: 5 μm; (**G**) proportion of cells with lagging chromosomes. HCT116 cells treated as in **F** were observed for the presence of lagging chromosomes. For each condition, 200 cells were observed. Experiments were repeated three times, and the average of each experimental result is shown as a dot. Error bars represent SD. * *p* < 0.05 (Student’s *t*-test); (**H**) micronuclei formation in cells depleted of KIF4A. HCT116 cells were treated as in (**F**) and interphase cells were observed. An arrow indicates a micronucleus. Scale bar: 5 μm; (**I**) proportion of cells with micronuclei. HCTT16 cells treated as in **F** were observed for the presence of micronuclei as shown in (**H**). For each condition, 1000 cells were observed. Experiments were repeated three times, and the average of each experimental result is shown as a dot. Error bars represent SD. * *p* < 0.05 (Student’s *t*-test).

**Figure 4 biomolecules-09-00010-f004:**
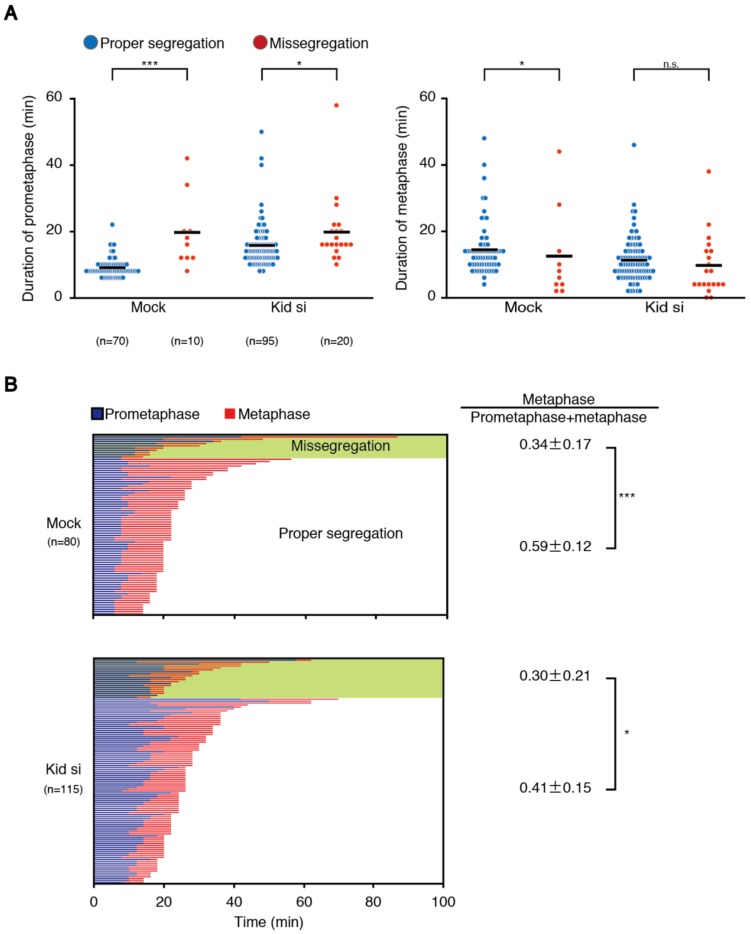
Cells that underwent chromosome missegregation exhibit an elongated prometaphase and a shortened metaphase. (**A**) Duration of prometaphase and metaphase depending on the occurrence of chromosome missegregation. HCT116 cells expressing H2B-mCherry were transfected with or without the siRNA for Kid (#1) and subjected to live cell imaging. Time from nuclear envelope breakdown to anaphase onset was measured in at least 80 mitotic cells for each condition, and categorized by the occurrence of chromosome missegregation. Number of cells observed is shown. The average is indicated with a bar. n.s., not statistically significant, * *p* < 0.05, *** *p* < 0.0005 (Mann-Whitney *U* test); (**B**) duration of prometaphase and metaphase in each cell in (**A**) is indicated with a bar, categorized by the occurrence of chromosome missegregation. Number of cells observed is shown. Ratio of metaphase time to the sum of prometaphase and metaphase time for each category is shown (mean ± SD). * *p* < 0.05, *** *p* < 0.0005 (Student’s *t*-test and Welch’s *t*-test).

**Figure 5 biomolecules-09-00010-f005:**
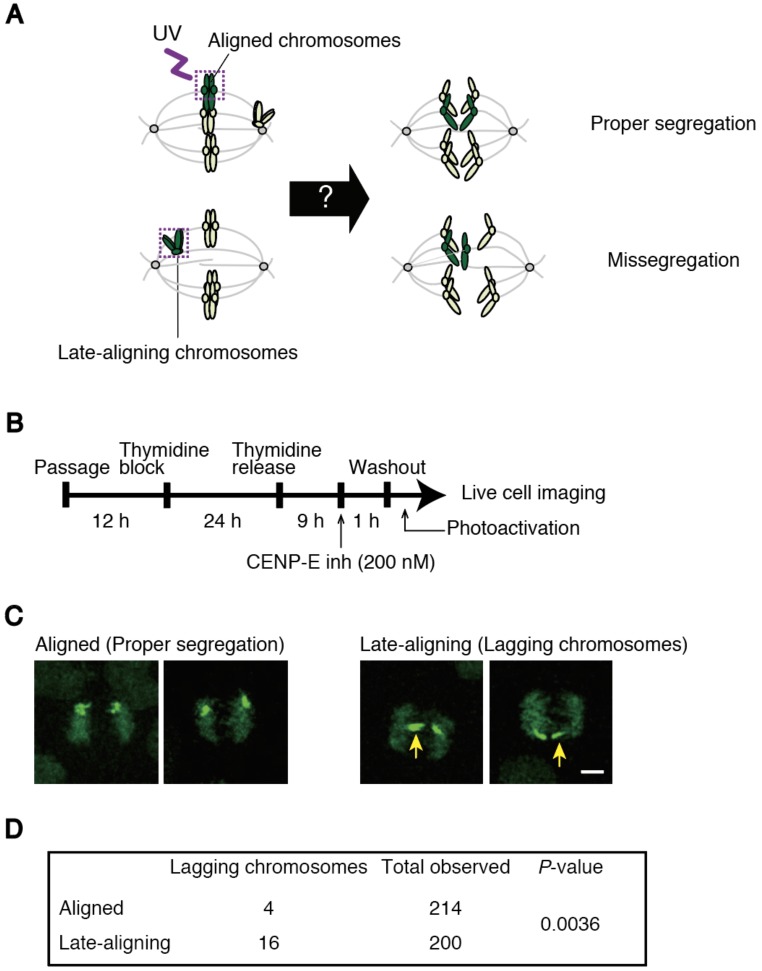
Late-aligning chromosomes are preferentially missegregated. (**A**) Schematic of experiment to track late-aligning chromosomes by photoactivation-based labeling; (**B**) experimental procedure to photoactivate chromosomes after washout of a CENP-E inhibitor; (**C**) examples of tracking of photoactivated chromosomes. Photoactivated aligned chromosomes that segregated properly (left), or late-aligning chromosomes that gave rise to lagging chromosomes (right) are shown. Lagging chromosomes are indicated by yellow arrows. Scale bar: 5 μm; (**D**) incidence of lagging chromosomes for aligned or late-aligning chromosomes. *P*-value was obtained using Chi-squared test.

**Figure 6 biomolecules-09-00010-f006:**
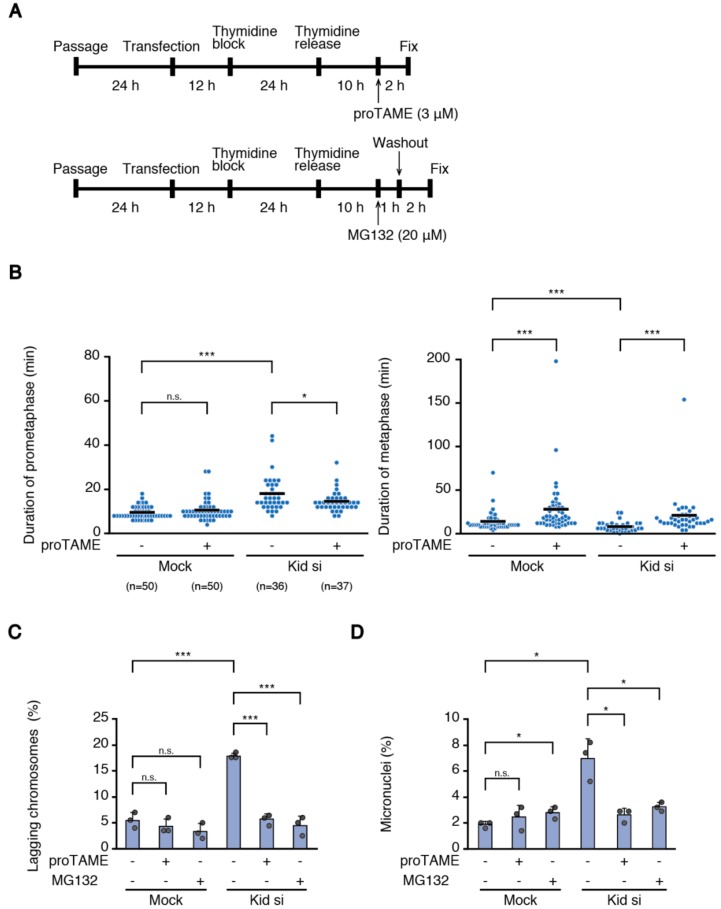
Delaying anaphase onset ameliorates chromosome missegregation in Kid-depleted cells. (**A**) Experimental procedure to treat cells with proTAME (upper), or MG132 (lower), after RNAi; (**B**) duration of prometaphase (left) or metaphase (right) in proTAME-treated cells. HCT116 cells expressing H2B-mCherry were transfected with or without the siRNA for Kid (#1), and treated with or without proTAME, according to the procedure shown in (**A**). Number of cells observed is shown. The average is indicated with a bar. n.s., not statistically significant, * *p* < 0.05, *** *p* < 0.0005 (Mann-Whitney *U* test); (**C**) proportion of lagging chromosomes in Kid-depleted cells treated with proTAME or MG132. For each condition, 200 cells were observed. Experiments were repeated three times, and the average of each experimental result is shown as a dot. Error bars represent SD. n.s., not statistically significant, *** *p* < 0.0005 (Student’s *t*-test); (**D**) proportion of micronuclei formation in Kid-depleted cells treated with proTAME or MG132. For each condition, 1000 cells were observed. Experiments were repeated three times, and the average of each experimental result is shown as a dot. Error bars represent SD. n.s., not statistically significant, * *p* < 0.05 (Student’s *t*-test and Welch’s *t*-test).

**Figure 7 biomolecules-09-00010-f007:**
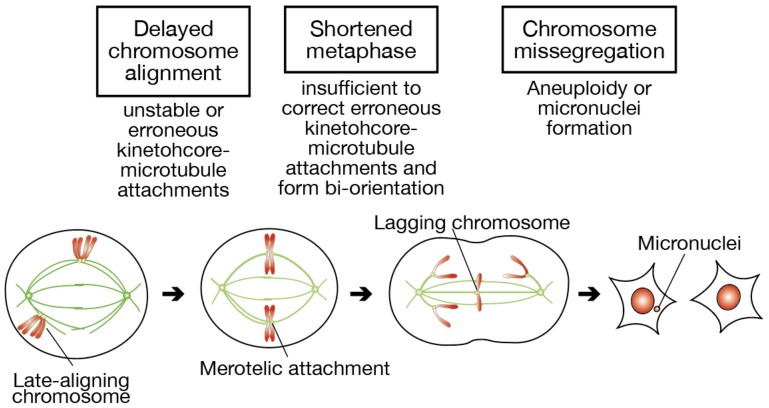
A schematic model depicting the relationship between delayed chromosome alignment and an increased rate of chromosome missegregation. When chromosome alignment is delayed, kinetochore-microtubule attachments of late-aligning chromosomes near spindle poles are unstable or might be erroneous. At the expense of elongated prometaphase, metaphase duration becomes shortened, which is insufficient for late-aligned chromosomes to correct erroneous kinetochore-microtubule attachments and form bi-orientation. Persistence of erroneous kinetochore-microtubule attachments at anaphase onset can result in chromosome missegregation such as the appearance of lagging chromosomes. A fraction of lagging chromosomes give rise to aneuploidy or micronuclei formation, causing genomic instability.

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
