# Peer review of "Delayed Chromosome Alignment to the Spindle Equator Increases the Rate of Chromosome Missegregation in Cancer Cell Lines"

_biomolecules, 2018, doi:10.3390/biom9010010_

Round 1
Reviewer 1 Report
The manuscript by Dr. Kozo Tanaka and colleagues describes the impact of slight delays in the alignment of chromosomes at the spindle equator on the rate of mitotic chromosome missegregation. This work addresses the role of chromokinesins Kid and KIF4A in directing the chromosomes away from the spindle poles, where Aurora A kinase (kinetochore-microtubule attachment destabilizer) is concentrated. To date, our understanding from chromosome alignment studies recognize the failure of timely congression of chromosomes at the spindle equator as a consequence of erroneous kinetochore-microtubule attachments, but this work suggests the importance of timely chromosome alignment in establishing proper kinetochore-microtubule attachments. The authors show that depletion of Kid and KIF4A motor proteins slightly yet significantly delays the chromosomes from aligning at the spindle equator, presumably due to difficulty in establishing amphitelic attachments, and increased the frequency of lagging chromosomes. By using photoactivatable GFP to mark chromosomes that are late to arrive at the metaphase plate, the authors were able to follow the same chromosomes during their later segregation and demonstrate that these chromosomes indeed are often missegregated. Finally, the authors show that increasing the duration of metaphase by delaying anaphase onset can alleviate the chromosome missegregation, presumably by allowing more time for lagging chromosomes to reach their respective positions in chromokinesins depleted cells.
I believe the results described in this manuscript are original and would be of significance for understanding chromosome segregation defects in somatic cells. In general, the data are solid and clearly and convincingly presented. To ensure reproducibility, the authors need include more methodological details for some experiments (Major comments). In addition, the authors may wish to consider a few additional changes for the sake of clarity (Minor comments).
Major comments:
1. For the experiments in Figure 6, in which proTAME and MG132 were used to delay anaphase onset, the authors need to include in the Materials and Methods section information about what solvent was used, what the stock and working concentrations were for each drug (currently, only the working concentration is given), and whether the control experiment included carrier in a working concentration equivalent to that used in the drug treatment.
2. In Figure 6, the authors use MG132 to investigate the effects of delayed anaphase onset on the fidelity of chromosome segregation. In Figures 2B, 2C, S2B and S2C, cells were also treated with MG132 prior to fixation. Presumably, this treatment was done to increase the number of metaphase figures available for analysis. The authors should state clearly in the Results section that this treatment was used and why, and they should include in their Discussion section a comment on whether the one hour treatment used in Figs 2 and S2 might have the same effect as the one hour treatment followed by two hour long washout in Fig 6.
Minor comments:
1. The formatting of several scatterplots in the text obscures the large number of observations in the underlying data. For example, in Figure 4A, 10 occurrences of missegregation were reported in the mock-treated control, but only 7 dots are apparent, with the remaining three presumably being mostly “behind” other dots. This issue is true of all the data in this figure and also in Supplemental Figure S3. Presumably, there is a setting in the software used to prepare these plots that can increase the lateral spacing between datapoints to better visualize the data. The results are clear in the current figures, but they would be even more convincing if the plots were allowed to “spread out” more.
2. In the first few sentences of the introduction, the authors may want to use more precise language. Bi-orientation is indeed required for accurate chromosome segregation during mitosis, but mono-orientation is required during meiosis I. Simply changing the initial sentence to, “For proper mitotic chromosome segregation…” would solve this problem. Similarly, some organisms including budding yeast undergo a closed mitosis in which the nuclear envelope never breaks down. Thus, the statement about attaching chromosomes to MTs requiring NEBD is true in many, but not all, cell types.
3. In general, the paper is clearly written. There are a few grammatical errors in the manuscript (including the abstract – “Cells that underwent chromosome missegregation…”). Although the intended meaning is nevertheless clear, a grammar check in Microsoft Word or at grammarly.com would likely identify many of these issues, should the authors desire to fix them.
Reviewer 2 Report
The manuscript by Kumiyasu addresses the question whether delayed chromosome alignment during congression in pro metaphase leads to missegration in anaphase. This is an important and timely question and the study will be of interest to a wide range of scientists investigating mechanisms of mitosis and tumorigenesis.
The authors follow the behaviour of chromosomes directly by time lapse analysis. Individual chromosomes are followed in that they label selected chromosomes by photo activation of a histone-GFP. Missegregation is observed by lagging chromosomes during anaphase and in chromosome spreads. Delayed chromosome alignment is induced by siRNA mediated depletion of two kinesins: Kid and KIF4A.
The authors report that Kid and KIF4A depleted cells are characterised first by a prolonged pro metaphase when the chromosomes congress at the metaphase plate and second by an increased chromosome missegregation (lagging chromosomes, number of micronuclei, chromosome number), Fig. 2, 3, S1, S2). Then the authors correlated these two phenotypes with each other (Fig. 4. 5, S3, S4). This is best shown in the assay in which selected chromosomes are labeled by photo activation. Here they report a weak correlation of delayed alignment during congression and missegregation in anaphase, but importantly they detect a statistically significant increase in cells CENP-E inhibited cells. Lastly, the authors show that during a lengthened metaphase can improve the missegregation defects caused by Kid depletion (Fig. 6).
I clearly recommend publication of the manuscript. The data are presented in an easily accessible manner. The conclusions are supported by the data. Before publication the following criticism and suggestions should be incorporated into a revised version of the manuscript. New experimental data is not required.
- At some points the author over generalise their conclusions. For example, the relation of delayed chromosome alignment and missegoatation is stated without restrictions. However, the data are based on experiments in tumour cells. In fact the authors state that Kid depletion in „normal“ cells (PRE-1) did not lead to significant increase in missegregation. The authors should avoid to give the impression that delayed alignment is linked very indirectly to missegregation, as the correlation is very low, only 4/16 out to 214/200 chromosomes with a delayed alignment were missegregated.
- the title should include a statement such as „… in tumour cells“.
- A large part of the supplemental data should be included into the main figures. The data about KIF4A depleted cells are primary data and should be shown in the figures parallel to the KIP data. The images (but not quantification) of both two Kid siRNAs are not informative, images of one siRNA is sufficient.
- Wherever possible, the individual data points should be shown, e. g. Fig 1C, 2B, 2D, 6C, 6D, S2F, S2H. If the number of data points is high, box plots may be used for presentation.
- two movies are included in the supplemental data showing behaviour of individually labelled chromosome. The Movie 2 shows a chromosome with a delayed alignment but a normal segregation. it would be informative to show an example of a chromosome with a delayed alignment and following missegregation.
- The data are largely based on time lapse imaging. More data should be represented in images series such as in Fig. 2D with several snap shots from the same representative movie. In Fig. 3A the behaviour of chromosomes in kid depleted cells could be shown in an image series (Fig. 3A), Similarly an image series would be informative in Fig. 5C, S1A, S2B, S2E, S4A
- No „rescue“ experiments with RNAi resistent constructs were conducted to demonstrate the specificity of the siRNAs. I do not want to ask for such experiments, but a statement in the text should be included about this problem. Given that the effects are only clearly detected in tumour cells but not normal cells, the genetic background seems to play an important role in the phenotypes reported.
- Statistical analysis, page 4: „…. when sample size was small….“ How was small defined?
- The discussion is rather long. A condensed version avoiding a too generalised discussion would be more appropriate to my taste.